# A Method for Detecting Outliers from the Gamma Distribution

**Xiou Liao \*, Tongtong Wang and Guohua Zou**

School of Mathematical Sciences, Capital Normal University, Beijing 100048, China
\*   Correspondence: 2220501018@cnu.edu.cn

**Abstract:** Outliers often occur during data collection, which could impact the result seriously and lead to a large inference error; therefore, it is important to detect outliers before data analysis. Gamma distribution is a popular distribution in statistics; this paper proposes a method for detecting multiple upper outliers from gamma $(m, \theta)$. For computing the critical value of the test statistic in our method, we derive the density function for the case of a single outlier and design two algorithms based on the Monte Carlo and the kernel density estimation for the case of multiple upper outliers. A simulation study shows that the test statistic proposed in this paper outperforms some common test statistics. Finally, we propose an improved testing method to reduce the impact of the swamping effect, which is demonstrated by real data analyses.

**Keywords:** outliers detecting; gamma distribution; critical values; swamping effect

**MSC:** 62F03

## 1. Introduction

The presence of outliers in the data may have an appreciable impact on the data analysis, which often leads to erroneous conclusions, and in turn results in severe decision-making mistakes. Therefore, it is necessary to detect outliers before statistical analysis. On the other hand, outlier detection has a wide range of applications in the prevention of financial fraud, disease diagnosis, and judgment of the truth of military information, etc.

Refs. [1,2] define outliers as those observations which are surprisingly far away from the main group. In a one dimensional situation, if the observations are arranged in an ascending order of magnitude, there will be only three types of outlier detection problems: (i) only upper outliers; (ii) only lower outliers; and (iii) both upper and lower outliers.

The commonly used methods of dealing with outliers include the detection of outliers and robust statistical methods. Robust methods aim to analyze data while retain outliers and minimize the deviation of analytical results from theoretical results. The detection of outliers is to identify outliers in the sample by using a reasonable statistical procedure and then analyzing the remaining observations. In this paper, we focus on this method.

In the field of statistics, there are many results on the detection of outliers, and many effective methods have been proposed. These methods include descriptive statistics, machine learning, and hypothesis testing.

Descriptive statistics is intuitive and contains no computational burden. Commonly used methods include Box-plot, Hampel rule, etc. Box-plot needs to compute the 3/4 quantile and 1/4 quantile of the sample, $Q_3$ and $Q_1$. Denote $IQR = Q_3 - Q_1$ as the interquartile range, then the observations are located in the interval of $[Q_1 - 1.5IQR, Q_3 + 1.5IQR]$ in the plot are observed as clean observations, and other observations are tested as outliers. According to [3], a data point is identified as an outlier if the distance between it and the sample median exceeds 4.5 times MAD, where $MAD(X) = med|X - med(X)|$.

Machine learning mainly trains the sample to detect outliers according to the data characteristics, combined with mathematical models and statistical principles. Some common methods include one-class support vector machines (one-class SVM), minimum spanning

tree (MST), etc. One-class SVM usually trains a minimal, ellipsoid which contains all normal observations from historical data or other clean data. Then, the observations that fall outside the ellipsoid are treated as outliers; see [4]. MST algorithm defines the distance between points as Euclidean distance, considers the points as nodes, and finds a path connecting each node with the smallest sum of distances. Then, based on the given criteria, the sample is divided into different classes. The largest set is treated as inlying data, while the rest is treated as outliers; see [5].

Hypothesis testing is a basic method for outlier detection. By setting appropriate null and alternative hypotheses and constructing test statistics with certain properties, the hypothesis testing method can detect whether there are outliers in the sample with the given significance level.

In a univariate sample, and unlike the limitations of the exponential distribution, observations from gamma distribution are more extensive and easier to collect. This paper studies the multiple outlier detection under gamma distribution, a parameter $\theta$ slippages model. Since the 1950s, there has been many results about outlier detection based on the hypothesis testing method, but most of them aim to detect a single outlier or outliers in a normal distribution. In the 1970s, outlier detection under more general distributions such as exponential, Pareto, and uniform distributions received much attention. Multiple outlierdetection has recently drawn considerable attention in practice owing to the development of science and technology and the diversification of data collection methods. We briefly introduce three commonly used statistics, which are suitable for detecting multiple upper outliers in the gamma distribution.

Dixon's statistic proposed in [6] is based on the idea that the dispersion of the suspect observations accounts for a large proportion of the sample dispersion. This method is further extended in [7–9], where [8] proposes the following statistic

$$D_k = \frac{X_{(n)} - X_{(n-k)}}{X_{(n)} - X_{(1)}}. \tag{1}$$

With the given significance level $\alpha$, $X_{(n-k+1)}, \cdots, X_{(n)}$ are identified as outliers if $D_k > d_k(\alpha)$, where $d_k(\alpha)$ is the critical value of $D_k$. Later, another Dixon type statistic for detecting outliers in a gamma distribution is proposed in [10,11], and the statistic is

$$L_k = \frac{X_{(n)} - X_{(n-k)}}{X_{(n)}}. \tag{2}$$

Ref. [10] gives the critical value $l_k(\alpha)$ for the given significance level $\alpha$, $X_{(n-k+1)}, \cdots,$ $X_{(n)}$ are regarded as outliers if $L_k < l_k(\alpha)$. The third test statistic is $N_k$ by [10,11]:

$$N_k = \frac{X_{(n-k)} - X_{(1)}}{\sum_{j=n-k+1}^{n}(X_{(j)} - X_{(1)})}. \tag{3}$$

Ref. [10] also obtains the corresponding critical value $n_k(\alpha)$ for the given significance level $\alpha$. $X_{(n-k+1)}, \cdots, X_{(n)}$ are regarded as outliers if $N_k < n_k(\alpha)$. The fourth test statistic is a "gap-test" ([12]), which is given by

$$Z_k = \frac{X_{(n)} - X_{(n-k)}}{\sum_{j=1}^{n} X_{(j)}}. \tag{4}$$

Ref. [12] provides the critical value $z_k(\alpha)$ for the significance level $\alpha$, and $X_{(n-k+1)},$ $\cdots, X_{(n)}$ are identified as outliers if $Z_k > z_k(\alpha)$. The fifth test statistic is proposed in [13], which is given by

$$V_k = \frac{\sum_{j=1}^{k}(X_{(n-k+j)} - X_{(n-k)})}{\sum_{j=2}^{n}(X_{(j)} - X_{(1)})}. \tag{5}$$

Ref. [13] shows that the distribution of $V_k$ and the critical value $v_k(\alpha)$ can be obtained for the given significance level $\alpha$. Thus, $X_{(n-k+1)}, \cdots, X_{(n)}$ are regarded as outliers if $V_k > v_k(\alpha)$.

The remainder of this article is organized as follows. In Section 2, we propose a test statistic to detect outliers in a gamma sample, and the density function of the proposed test statistic is derived. In order to obtain the critical values, a Monte Carlo procedure and a kernel density estimation procedure are proposed. In Section 3, the simulation results demonstrate that the proposed $T_k$ test statistic is better than others. Furthermore, an improved $T_k$ method is suggested, which can eliminate the swamping effect in multiple outliers detection in Section 4. A real data analysis is performed in Section 5. Section 6 is the conclusion. All proofs of theoretical results are presented in Appendix A, and the data of empirical applications is contained in Appendix B.

## 2. Model Framework and Methodology for Detecting Outliers

In this section, we propose a testing method to detect upper outliers from a gamma distribution. Both single and multiple outliers are considered. We will derive the distribution of the test statistic $T_k$ for single upper outlier detection, and design two methods—the Monte Carlo method and the kernel density method—to calculate the critical value of $T_k$ for multiple outliers.

### 2.1. Model Framework

Assume the null distribution is gamma distribution, gamma $(m, \theta)$, with the density function given by

$$f(x|m, \theta) = \frac{\theta^m}{\Gamma(m)} x^{m-1} e^{-\theta x}, x > 0, \tag{6}$$

where $m$ and $\theta$ are unknown, $m, \theta > 0$. The null hypothesis is

$$H : X_1, \cdots, X_n \sim f(x|m, \theta).$$

Then, the density function in the alternative hypothesis is

$$f(x|m, \theta, \lambda) = \frac{(\lambda\theta)^m}{\Gamma(m)} x^{m-1} e^{-\lambda\theta x}, x > 0, 0 < \lambda \leq 1, \tag{7}$$

where $\lambda$ denotes the contaminant factor. The slippage alternative hypothesis is

$$\bar{H} : n - k \text{ observations} \sim f(x|m, \theta), \text{and } k \text{ observations} \sim f(x|m, \theta, \lambda).$$

Sorting $X_1, \cdots, X_n$ from small to large, we obtain the sample $S = X_{(1)}, \cdots, X_{(n)}$, where $X_{(j)}$ corresponds to the $j$th observation in $S$. When $k = 1$, $X_{(n)}$ is the suspicious point, we propose the test statistic $T_{(n)}$ to detect an outlier in $S$,

$$T_{(n)} = \frac{X_{(n)}}{\bar{X}}. \tag{8}$$

For a given significance level $\alpha$, letting $t_1(\alpha)$ be the critical value, and $X_{(n)}$ is detected as an outlier if $T_{(n)} > t_1(\alpha)$. When $k > 1$, we propose the following test statistic to detect multiple outliers,

$$T_k = \frac{\sum_{j=n-k+1}^{n} X_{(j)}}{\bar{X}}. \tag{9}$$

For a given significance level $\alpha$, if we let $t_k(\alpha)$ be the critical value, $X_{(n-k+1)}, \cdots, X_{(n)}$ are detected as outliers if $T_k > t_k(\alpha)$.

**Theorem 1.** *$T_k$ is a test statistic that is derived from the likelihood ratio principle.*

**Proof of Theorem 1.** See Appendix A.1. □

*2.2. Detecting Single Outlier*

$T_{(n)}$ can be used for testing a single upper outlier for the gamma sample. To obtain the critical value of the test, we derive the distribution of $T_{(n)}$ under the null model, as follows.

Denote $T_j = T_{n,j} = \frac{X_j}{\bar{X}}$ and $T_{(j)} = T_{n,(j)} = \frac{X_{(j)}}{\bar{X}}$. Note that $X_1, X_2, \cdots, X_n$ are independent, so $\frac{X_j}{\sum_{j=1}^{n} X_j}$ follows beta $(m, (n-1)m)$ under the null model. Let $a = m$ and $b = (n-1)m$, for any $j$, the density function of $\frac{X_j}{\sum_{i=1}^{n} X_j}$ is

$$\beta_{a,b}(u) = \{\Gamma(a+b)/\Gamma(a)\Gamma(b)\}u^{a-1}(1-u)^{b-1}, 0 < u < 1. \tag{10}$$

As $T_j = T_{n,j} = \frac{X_j}{\bar{X}} = n\frac{X_j}{\sum_{i=1}^{n} X_i}$, the density function of $T_j$ is given by

$$\beta_{a,b}(v) = \frac{\{\Gamma(a+b)/\Gamma(a)\Gamma(b)\}v^{a-1}(n-v)^{b-1}}{n^{a+b-1}}, 1 < v < n. \tag{11}$$

It can be shown that

**Lemma 1.** *Assume that $X_1, \cdots, X_{n-1}, X_n$ are independent identically from gamma $(m, \theta)$, then $\frac{\max\limits_{k \neq n} X_k}{\sum_{j=1}^{n-1} X_j}$ and $\frac{X_n}{\sum_{j=1}^{n-1} X_j}$ are independent.*

**Proof of Lemma 1.** See Appendix A.1. □

**Theorem 2.** *If $X_1, X_2, \cdots, X_{n-1}, X_n$ are independent from gamma $(m, \theta)$, then the density function of $T = \frac{X_{(n)}}{\bar{X}}$ is*

$$n\beta_{m,(n-1)m}(v)A_{n-1}[(n-1)\frac{v}{n-v}], 1 < v < n, \tag{12}$$

*where $A_n(v)$ is the cumulative distribution function (CDF) of $T_{n,(n)}$.*

**Proof of Theorem 2.** See Appendix A.1. □

The density function of $T_{(n)} = \frac{X_{(n)}}{\bar{X}}$ under the null model is an iterative function and the critical value of $T_{(n)}$ can be obtained by Equation (12).

*2.3. Detecting Multiple Outliers*

$T_k$ with $k > 1$ can be used to detect outliers in the gamma sample if there exist multiple outliers. However, deriving the distribution of $T_k$ is a difficult task. In this case, to obtain the critical value of the test, we propose two methods, the Monte Carlo method and the kernel density estimation method.

2.3.1. Monte Carlo Method

First, note that the distribution of $X_{(j)}/\bar{X}$ is unrelated to $\theta$ under the null model. Based on this property, the Monte Carlo method for computing the critical value of the $T_k$ test is given below.

Parameter $m$ can be obtained by the Newton-Rapson algorithm which is based on the sample or estimated by other samples, empirical methods, and so on. We consider the outliers from a slippage model in which the parameter $\theta$ has been shifted to $\lambda\theta$, with the parameter $m$ being fixed, where $0 < \lambda \leq 1$ is the contamination factor.

The idea of the Monte Carlo method is generating $n$ samples, and $T_K$ can be obtained from each sample. Denote $S_M$ as the set that consists of all $T_k$. Then, based on the law of

large numbers, we use the $1 - \alpha$ quantile of $S_M$ as the estimate of $t_k(\alpha)$. The pseudocode of the Monte Carlo method is given by Algorithm 1.

---

**Algorithm 1** Monte Carlo method

---

**Input:** Parameters

　$n$: sample size;

　$k$: number of suspicious observations;

　$\alpha$: the significance level, say, $\alpha = 0.05$;

　$u$: number of samples, say, $u = 5000$.

**Output:** $t_k(\alpha)$.

　**for** $j$ in $1 : u$ **do**

　　generate $n$ observations from gamma$(m, 1)$;

　　$T_{k,j} = \frac{\sum_{i=n-k+1}^{n} X_{(j,i)}}{\bar{X}_j}$;

　**end for**;

　get $S_M = \{T_{k,1}, \cdots, T_{k,u}\}$;

　$t_k(\alpha) \leftarrow (1 - \alpha)$ quantile of $S_M$.

---

Using the above Monte Carlo method to compute the critical values of the $T_k$ test statistic for different $n$, $k$, and $m = 5$, the results are summarized in Table 1.

**Table 1.** The critical values of $T_k$ in the case of $m = 5$ and significance level $\alpha = 0.05$.

| $k$ ╲ $n$ | 100 | 120 | 150 | 200 |
|---|---|---|---|---|
| 10 | 20.85 | 21.43 | 22.15 | 23.08 |
| 20 | 35.78 | 37.10 | 38.65 | 40.79 |
| 30 | 48.49 | 50.54 | 53.06 | 56.19 |
| 40 | 59.50 | 62.49 | 66.06 | 70.36 |
| 50 | 69.29 | 73.23 | 77.91 | 80.50 |

2.3.2. Kernel Density Estimation Method

This method aims to use a large sample of $T_k$ to approach its density function, and the estimated function is denoted as $f(x)$. Then, with the significance level $\alpha$, we compute $t_k(\alpha)$ from

$$\int_{t_k(\alpha)}^{+\infty} f(x)dx = \alpha. \tag{13}$$

Using a Gaussian kernel function, we have

$$K(\frac{x - x_j}{h}) = \frac{1}{\sqrt{2\pi}} e^{-\frac{(x-x_j)^2}{2h^2}}, \tag{14}$$

where $x_j = T_{k,[j]}$ and $h$ is the bandwidth. Therefore, the estimated density function of $T_k$ is

$$f(x) = \frac{1}{uh} \sum_{j=1}^{u} \frac{1}{\sqrt{2\pi}} e^{-\frac{(x-x_j)^2}{2h^2}}. \tag{15}$$

The pseudocode of the kernel density estimation method is given by Algorithm 2.

---

**Algorithm 2** Kernel density estimation method

---

**Input:** Parameters

$n$: sample size;

$k$: number of suspicious observations;

$\alpha$: the significance level, say, $\alpha = 0.05$;

$u$: number of samples, say, $u = 5000$.

**Output:** $t_k(\alpha)$.

**for** $j$ in $1 : u$ **do**

generate $n$ observations from gamma$(m, 1)$;

$T_{k,j} = \frac{\sum_{i=n-k+1}^{n} X_{(j,i)}}{\bar{X}_j}$;

**end for**;

get $S_M = \{T_{k,1}, \cdots, T_{k,u}\}$;

compute the bandwidth of $S_M$;

choose Gaussian kernel function, $K(\frac{x-x_j}{h}) = \frac{1}{\sqrt{2\pi}} e^{-\frac{(x-x_j)^2}{2h^2}}$, and the estimated density

function of $T_k$ is $f(x) = \frac{1}{uh} \sum_{j=1}^{u} \frac{1}{\sqrt{2\pi}} e^{-\frac{(x-x_j)^2}{2h^2}}$;

$t_k(\alpha) \leftarrow$ root of $\int_{t_k(\alpha)}^{+\infty} f(x)dx - \alpha = 0$.

---

Table 2 includes critical values of the $T_k$ test statistic for different $n$ and $k$ with $m = 5$ and $\alpha = 0.05$, which are calculated by the kernel density algorithm.

**Table 2.** The critical values of $T_k$ in the case of $m = 5$ and significance level $\alpha = 0.05$.

| $k$ \ $n$ | 100 | 120 | 150 | 200 |
|---|---|---|---|---|
| 10 | 20.81 | 21.44 | 22.12 | 23.09 |
| 20 | 35.79 | 37.12 | 38.79 | 40.81 |
| 30 | 48.53 | 50.61 | 53.14 | 56.28 |
| 40 | 59.52 | 62.57 | 66.18 | 70.44 |
| 50 | 69.29 | 73.26 | 77.98 | 80.51 |

After comparing a large number of simulation results of the Monte Carlo method and the kernel density estimation method, we find the difference of results between these two methods is very small. Therefore, which method is chosen depends on your personal preference.

More generally, Algorithms 1 and 2 contribute two feasible methods to calculate the critical values of any test statistics for the given significance level, sample size $n$, and presupposed $k$.

## 3. Simulation Study

In this section, we evaluate, by a simulation study, the performance of the proposed test statistic $T_k$ and compare it with the commonly used methods including $D_k$, $L_k$, $N_k$, $Z_k$, and $V_k$ given in Section 1.

### 3.1. Simulation Setting

To evaluate the performance of a test statistic in the outlier detection, we consider two cases with and without outliers. For the former, a test statistic can be evaluated by computing the power when there exist $k$ outliers in the gamma $(m, \theta)$, and the probability of its power is replaced by the frequency of identifying outliers correctly; for the latter, a test statistic can be evaluated by counting the number of times that inlying observations are misjudged as outliers, which is called "false alarm". A test statistic is better if it has higher power and lower "false alarm".

To use the similar simulation setting as in [10,12,13], we transform the $\lambda$ and $\theta$ in Equations (6) and (7) to $\frac{1}{\lambda}$ and $\frac{1}{\theta}$, respectively.

For computing the power, we generate $n$ observations from Equation (6) and sort these points from small to large. $X_{(n-k+1)}, \cdots, X_{(n)}$ are replaced by $\lambda X_{(n-k+1)}, \cdots, \lambda X_{(n)}$, which has the same effect as producing $k$ upper outliers from Equation (7). Where $k = 2, 5$, $\lambda$ in [1:2] (0.055), $n = 20$ and $m = 5$. To measure the "false alarm", denote $k_o$ as the number of outliers in the $k$ largest observations. When $k = 2$ (5), we have $k_o = 1$ (2). Generate $n - k_o$ observations from Equation (6), and generate $k_o$ from Equation (7). Then, detect the largest $k$ observations by using the different test statistics. These two cases with significance levels $\alpha = 0.01$ and 0.05. Our simulation study is carried out based on 2000 replications.

*3.2. Results*

For the case of outliers existing, the simulation results on the power of six test statistics are shown in Figures 1 and 2. It can be observed from Figure 1 that when $m = 5$, $k = 2$ and $\alpha = 0.01$, our test statistic $T_k$ has a higher power than the other five test statistics for the values of $\lambda$ smaller than 1.650; and for larger $\lambda$, $T_k$ is worse than $N_k$ and $V_k$ but better than $Z_k$, $D_k$, and $L_k$. For $\alpha = 0.05$, $T_k$ is worse than $N_k$ and $V_k$ but better than $Z_k$, $D_k$, and $L_k$. It is clear from Figure 2 that when $m = 5$ and $k = 5$, $T_k$ has the highest outlier detection capability for $\alpha = 0.01$; and if $\alpha = 0.05$, $T_k$ has the highest power for almost all the $\lambda$ values.

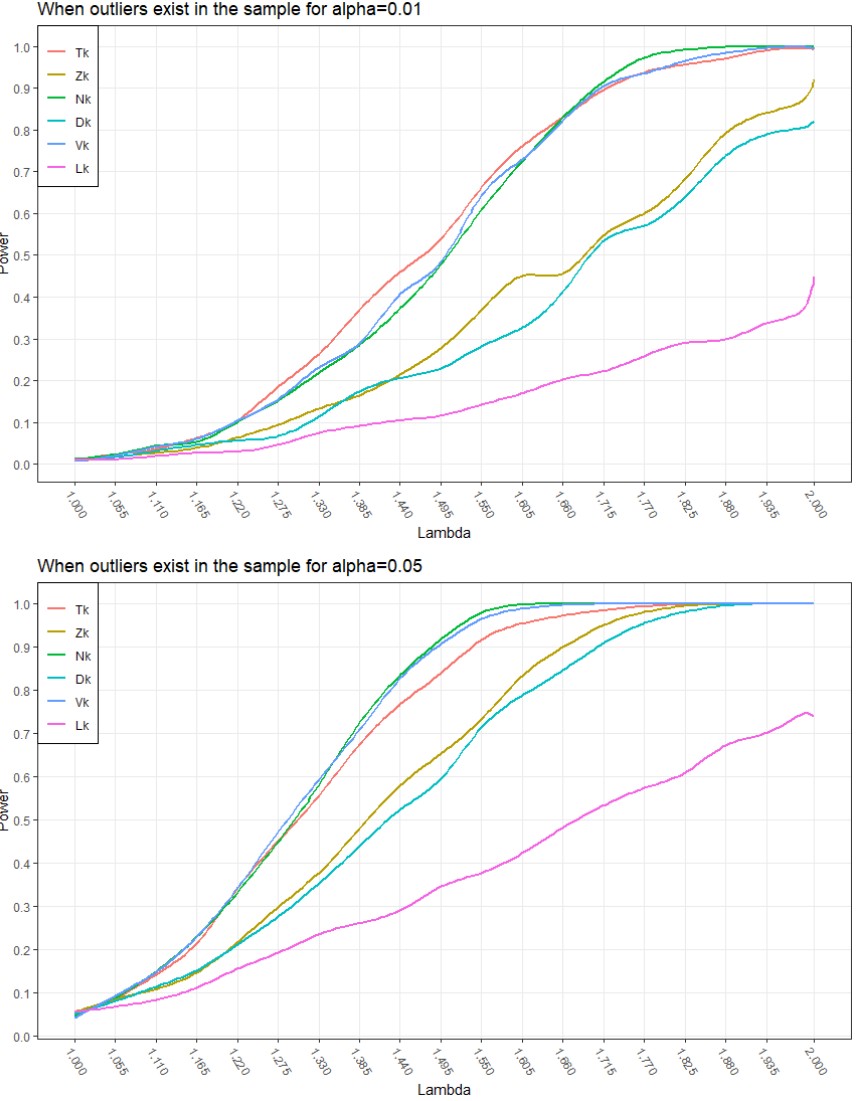

**Figure 1.** Power of test statistics for $m = 5$, $k = 2$, and $n = 20$.

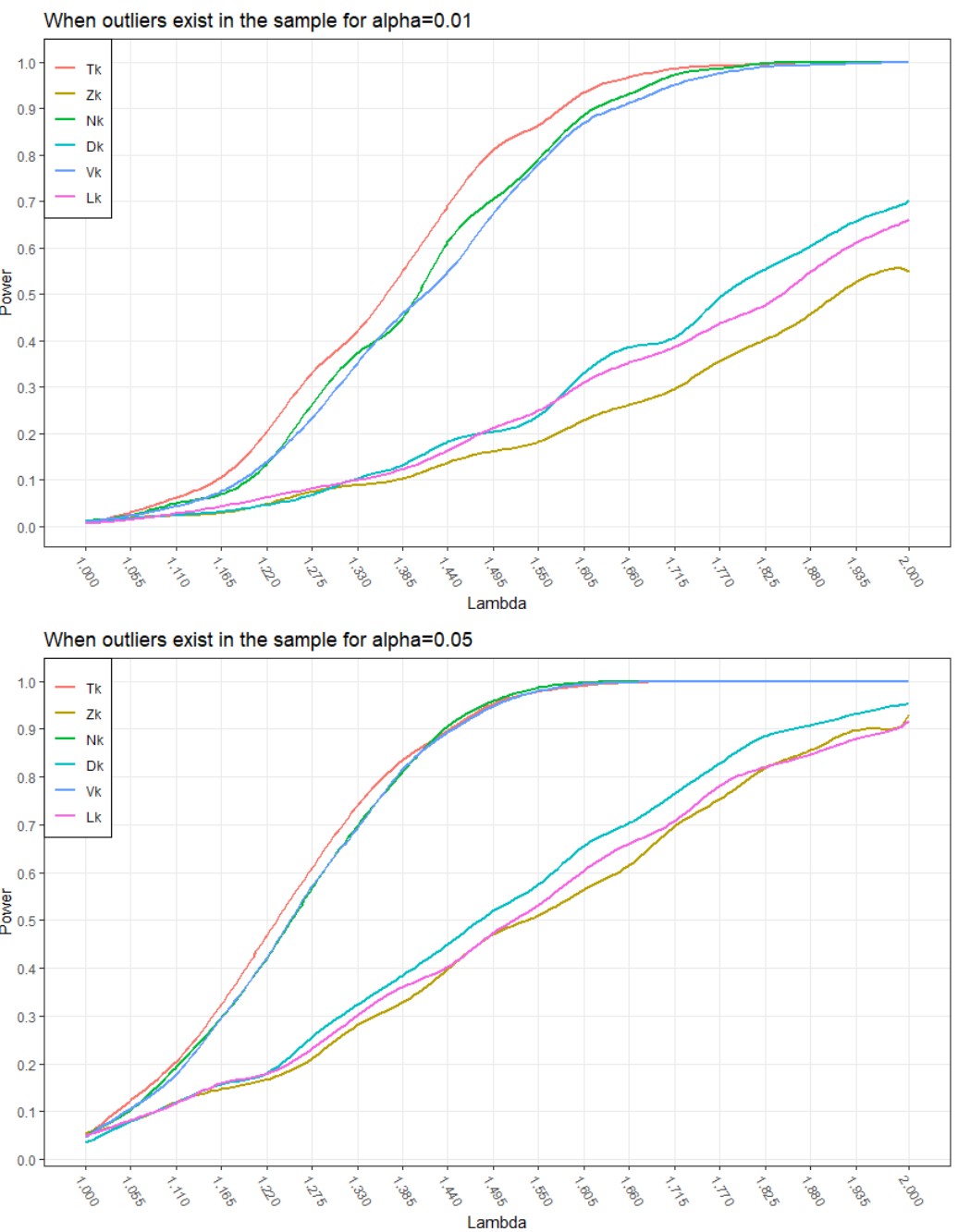

**Figure 2.** Power of test statistics for $m = 5$, $k = 5$, and $n = 20$.

For the case of the $k$ largest observations consisting of contaminants and some good observations, the simulation results on the swamping effect of six test statistics are shown in Figures 3 and 4. It can be observed from Figure 3 that for $k_o = 1$, with the significance level of 0.01, $T_k$ is better than $Z_k$ and $D_k$ but worse than $N_k$, $V_k$, and $L_k$. For $\alpha = 0.05$, the "false alarm" of $T_k$ is worse than that of $L_k$, but better than those of $Z_k$, $N_k$, $D_k$, and $V_k$. It is clear that the results of Figure 3 with $k_o = 1$ and Figure 4 with $k_o = 2$ are similar when $\alpha = 0.01$. For $\alpha = 0.05$, Figure 4 shows that $T_k$ is worse than $N_k$ and $V_k$ but better than $Z_k$, $D_k$, and $L_k$.

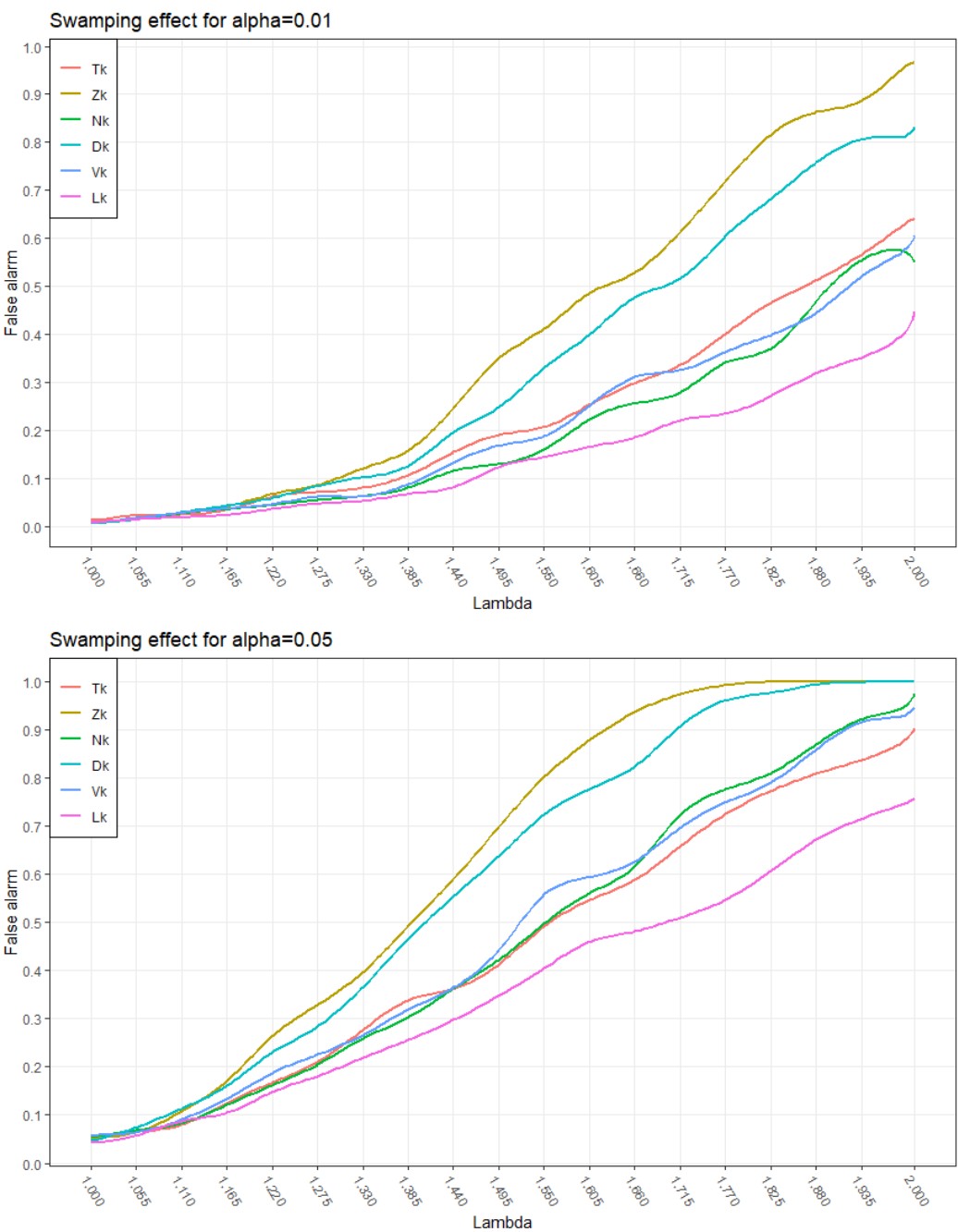

**Figure 3.** False alarm of statistics for $m = 5$, $k = 2$, $k_o = 1$, and $n = 20$.

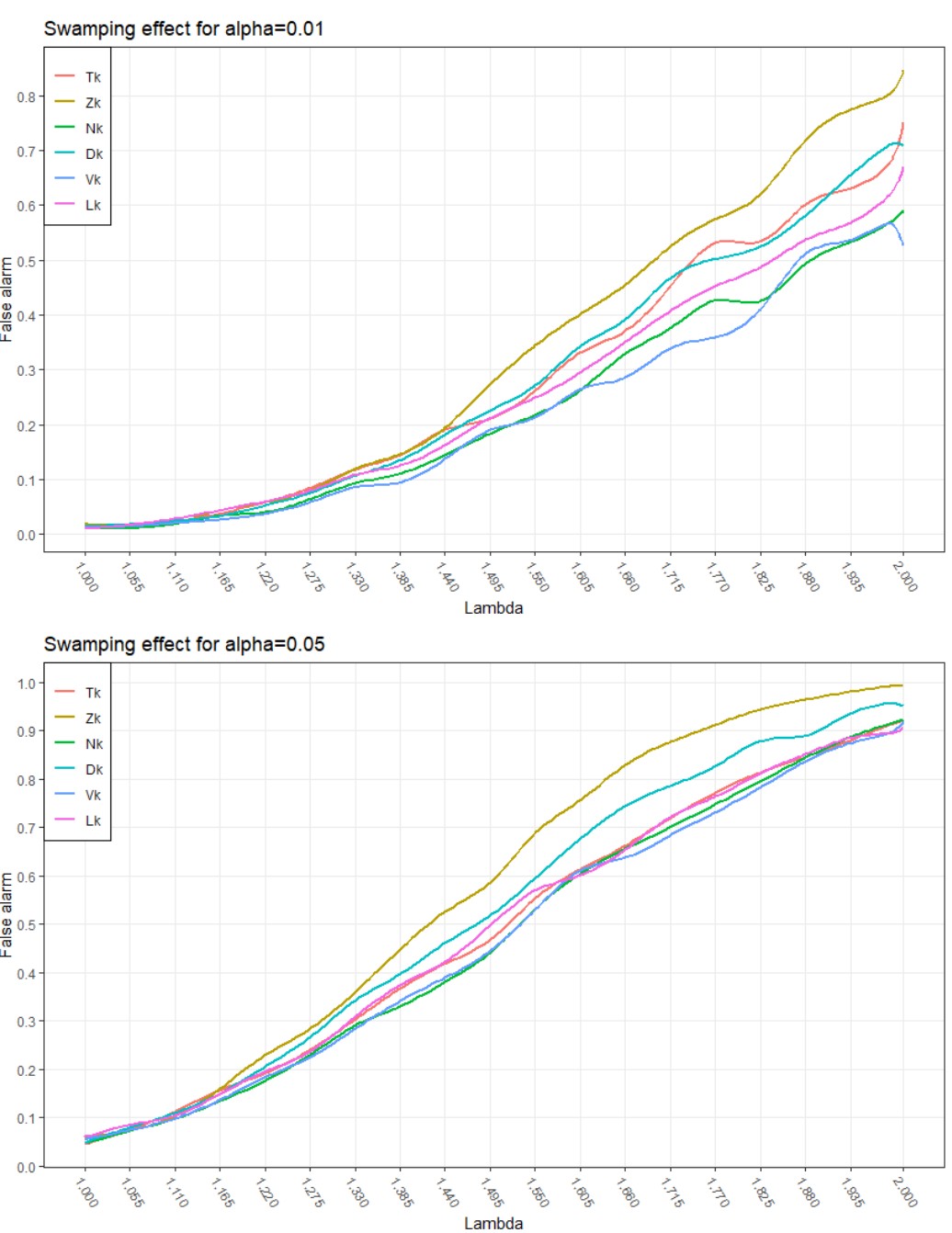

**Figure 4.** False alarm of statistics for $m = 5$, $k = 5$, $k_o = 2$, and $n = 20$.

In summary, the simulation results show that $T_k$ has the highest power and relatively lower "false alarm" than $Z_k$, $D_k$ and $L_k$ for $\alpha = 0.05$ and $k = 5$. With $k = 5$ and $\alpha = 0.01$, $T_k$ has the highest power than other test statistics, but the "false alarm" of $T_k$ is worse than those of $N_k$, $V_k$ and $L_K$. Therefore, with large $m$ and $k$, $T_k$ is generally better than $Z_k$, $N_k$, $D_k$, $V_k$, and $L_k$ for multiple outlierdetection.

## 4. Modified $T_k$ Test-ITK

In practice, almost all test statistics used to detect multiple outliers have the swamping effect. This phenomenon happens because large outliers may cause the sum of multiple observations to be too large in the block test. To reduce or eliminate the impact of the swamping effect, we suggest a modified $T_k$ test, ITK, which retains the high probabilities of outliers detecting and low error probabilities when there is no outlier in the gamma sample.

Note that for multiple outlier detection, some inlying observations may be judged as outliers falsely caused by improper $k$. For example, consider a sample consisting of $0.30, 0.62, 0.72, 0.80, 1.13, 1.42, 1.45, 2.30, 14.86$, and $22.01$, and use $T_3$ to test $X_{(10)} = 22.01$, $X_{(9)} = 14.86$, $X_{(8)} = 2.30$. Clearly, $T_3 = \frac{X_{(10)} + X_{(9)} + X_{(8)}}{\bar{X}} = 8.08$, and the critical value of $T_3$, by using Algorithm 1 in Section 2.3.1, is $t_k(0.05) = 6.43$. Therefore, $T_3 > t_k(0.05)$, and $X_{(10)}$, $X_{(9)}$, $X_{(8)}$ are outliers in the sample. However, in fact, $X_{(8)}$ is a genuineobservation from the inlying cluster. $X_{(8)}$ is detected as an outlier because $X_{(10)}$ and $X_{(9)}$ compared with the inlying sample are too large, causing the sum of $X_{(10)}$, $X_{(9)}$, $X_{(8)}$ beyond the bound range, i.e., swamping effect. However, this negative impact will be eliminated if we take $k = 2$.

To deal with the swamping effect, a method for choosing a reasonable $k$ should be given. Thus, our modified test includes two stages: (1) pick a reasonable $k$, and use the $T_k$ test to detect $k$ upper observations; (2) use stepwise forward testing for the remainingobservations or stepwise backward testing for the "outliers" sample from the first stage.

### 4.1. Estimation of k

From [14], the number of outliers should be less than $n/2$. Later, [1] put forward a point that the number of outliers is usually less than $\sqrt{n}$ if the sample is collected properly.

Here, we take

$$k = \hat{k} = [\sqrt{n}], \tag{16}$$

where $[\sqrt{n}]$ is the greatest integer less than or equal to $\sqrt{n}$.

### 4.2. The Improvement of the $T_k$ Test-ITK

Based on Section 4.1, we propose an improved $T_k$ test procedure, as follows:

Step 1. For the significance level $\alpha$, $X_{(n-k+1)}, \cdots, X_{(n)}$ are judged as outliers preliminarily, which forms a preliminary outliers sample, if $T_k > t_k(\alpha)$; otherwise, goto Step 5. The remaining observations constitute the preliminary inlying group, $S'$.

Step 2 (step forward test). Using step forward test to detect whether $S'$ includes any outliers. For $\alpha = 0.05$, $T_{[-k],1} = \frac{X_{(n-k)}}{\bar{X}_{[-k]}}$, and $X_{(n-k)}$ is an outlier if $T_{[-k],1} > t_{[-k],1}(\alpha)$; otherwise, goto Step 4.

Step 3. Repeat the test process in Step 2 until no outlier can be detected in $S'$. If $X_{(j)}$ is the smallest outlier in $S'$, then $X_{(j)}, \cdots, X_{(n)}$ are outliers in the data and stop the procedure.

Step 4 (step backward test). After the step forward test has stopped, use the step backward test to check the preliminary outliers sample in Step 1. For the significance level $\alpha = 0.05$, if $T_{[-k+1],1} = \frac{X_{(n-k+1)}}{\bar{X}_{[-k+1]}} > t_{[-k+1],1}(\alpha)$, then the step backward test ends; otherwise, use the step backward test for detecting $X_{(n-k+2)}$. Repeat this step until an outlier is detected. If $X_{(n)}$ is not judged as an outlier, then there is no outlier in the sample, the sample is inlying data.

Step 5. Let $\hat{k}_{new} = [\frac{\hat{k}}{2}]$, and substitute $k = \hat{k} = \hat{k}_{new}$ to Step 1. If $\hat{k}_{new} = 0$, there is no outlier in the sample, and the test procedure ends.

## 5. Empirical Applications

In this section, we apply the ITK test method to two data sets: Alcohol-related mortality rates and artificial scout position data, and compare it with the other six test statistics of $T_k$, $D_k$, $N_k$, $Z_k$, $V_k$, and $L_k$.

### 5.1. Alcohol-Related Mortality Rates in Selected Countries in 2000

The dataset (see Appendix B) is selected from Office for National Statistics (ONS). The Kolmogorov-Smirnov test indicates that this data follows the gamma distribution.

Here, $n = 100$ and so $\hat{k} = 10$. We obtain $m = 1.2$ by using the Newton-Rapson algorithm. From Appendix B, it is observed that $T_{10} = \frac{\sum_{j=91}^{100} X_{(j)}}{\bar{X}} = \frac{117.65}{2.47} = 47.70$. Further, we compute the critical value of $T_{10}$ by using Algorithm 1 in Section 2.3.1, and obtain

$t_{10}(0.05) = 21.99$. Obviously, $T_{10} > t_{10}(0.05)$, and hence $X_{(91)}, X_{(92)}, \cdots, X_{(100)}$ are detected as outliers preliminarily.

Then, we use the step forward test for the remaining sample. It is clear that $T_{(90)} = \frac{X_{(90)}}{\overline{X}_{[-10]}} = \frac{6.17}{1.43} = 4.31 < t_{[-10],1}(0.05) = 6.45$. Thus, $X_{(90)} = 6.17$ is a normal observation.

We now use the step backward test for $X_{(91)}, X_{(92)}, \cdots, X_{(100)}$. It is readily observed that $X_{(91)} = 10.17$, $T_{[-9],1} = \frac{X_{(91)}}{\overline{X}_{[-9]}} = \frac{10.17}{1.53} = 6.65$, and in the 5% significance level, $T_{[-9],1} = 6.55$. As $T_{[-9],1} > t_{[-9],1}(0.05)$, $X_{(91)}, X_{(92)}, \cdots, X_{(100)}$ are detected as upper outliers.

On the other hand, we utilize the $T_k$, $D_k$, $N_k$, $Z_k$, $V_k$, and $L_k$ test statistics to detect outliers, and the results are shown in Table 3.

**Table 3.** The outlier detection results of alcohol-related mortality rates by using various tests.

| Test Statistic | Number of Identified Observations | Number of Potentially Misjudged Observations |
|---|---|---|
| ITK | 10 | 0 |
| $T_k$ | 10 | 0 |
| $D_k$ | 0 | 0 |
| $N_k$ | 10 | 0 |
| $Z_k$ | 0 | 0 |
| $V_k$ | 10 | 0 |
| $L_k$ | 0 | 0 |

As we can observe from Table 3, ITK, $T_k$, $N_k$, and $V_k$ can identify outliers correctly without misjudgment. This phenomenon happens because $k$ is chosen reasonably. We can also observe that $D_k$, $Z_k$, and $L_k$ have bad performance in multiple upper outlier detection.

Furthermore, the result from Table 3 shows that Ireland, France, Austria, Slovenia, Portugal, Denmark, the United Kingdom of Great Britain and Northern Ireland, the Republic of Korea, the Russian Federation, and Australia have higher alcohol-related mortality rates, which means that these countries need to pay more attention to alcohol-related mortality.

### 5.2. Artificial Scout Position Data

In the application of military information, the gamma model is usually used to describe the position of some objects. Suppose a military scene, in a mission, 20 scouts reconnoiter a certain area, and their location components are characterized by $X_j$, $j = 1, \cdots, 20$, and the larger $X_j$, the further they are away from the landing site. If $X_j$ deviates from the main group, this indicates that the $i$th soldier is separated from the troops and may not be able to obtain support in time in case of an emergency. Therefore, it is necessary to pay attention to this movement.

In our setting, the basic model is gamma (3,5) and the alternative model is gamma (3,10). The initial data are outlined in Appendix B.

Here, $m = 3$ is known. The sample size is 20, thus $\hat{k} = [\sqrt{n}] = [\sqrt{20}] = 4$. From Appendix B, it is observed that $T_4 = \frac{\sum_{j=17}^{20} X_{(j)}}{\overline{X}} = \frac{0.91 + 2.90 + 3.32 + 3.44}{0.97} = 10.89$. With the significance level of 0.05, we utilize the Monte Carlo method to calculate the critical value for $T_4$, and we obtain $t_4(0.05) = 8.71$. As $T_4 > t_4(0.05)$, $X_{(17)}, X_{(18)}, X_{(19)}$ and $X_{(20)}$ are placed into the initial outlier group.

Furthermore, we continue to test the remained sample and carry out the step forward test for $X_{(16)} = 0.88$. Noting that $t_{[-4],1}(0.05) = 3.09 > T_{[-4],1} = \frac{X_{(16)}}{\overline{X}_{[-4]}} = \frac{0.88}{0.55} = 1.59$, $X_{(16)} = 0.88$ is not an outlier.

Presently, we use step backward test for $X_{(17)} = 0.91$, $X_{(18)} = 2.90$, $X_{(19)} = 3.32$, $X_{(20)} = 3.44$. It is clear that $T_{[-3],1} = \frac{X_{(17)}}{\overline{X}_{[-3]}} = \frac{0.91}{0.57} = 1.60$, and with the significance level of 0.05, $t_{[-3],1}(0.05) = 3.14$. Noting that $T_{[-3],1} < t_{[-3],1}(0.05)$, $X_{(17)} = 0.91$ is not an outlier.

Moreover, $T_{[-2],1} = \frac{X_{(18)}}{X_{[-2]}} = \frac{2.90}{0.70} = 4.13 > t_{[-2],1}(0.05) = 3.18$, the test procedure ends. Therefore, $X_{(18)} = 2.90$, $X_{(19)} = 3.32$ and $X_{(20)} = 3.44$ are outliers in the sample.

Meanwhile, we utilize the $T_k$, $D_k$, $N_k$, $Z_k$, $V_k$, and $L_k$ test statistics to detect outliers, and the results are shown in Table 4.

**Table 4.** The outlier detection results of artificial scout position data.

| Test Statistic | Number of Identified Observations | Number of Misjudged Observations |
|---|---|---|
| ITK | 3 | 0 |
| $T_k$ | 4 | 1 |
| $D_k$ | 4 | 1 |
| $N_k$ | 4 | 1 |
| $Z_k$ | 4 | 1 |
| $V_k$ | 4 | 1 |
| $L_k$ | 0 | 0 |

It can be observed from Table 4 that the ITK method performs better than the other five methods (the $T_k$, $D_k$, $N_k$, $Z_k$, $V_k$, and $L_k$ test statistics) because it can not only detect all outliers in the sample, but also has the lowest misjudged probabilities.

Further, from the result of the ITK method, we can obtain information that the IDs 18, 19, and 20 seem to be far away from the landing site. This means that they would be endangered in case of an emergency.

## 6. Concluding Remarks

It can be observed from the simulation that with the increase in $k$ and $n$ values, compared with other test statistics, our test statistic $T_k$ has a higher power and relatively lower "false alarm" on outlier detection, especially for a lower significance level. However, the swamping effect still exists for $T_k$, and this phenomenon will cause the loss of information. Therefore, to reduce the impact of swamping effect, we design the ITK test. From the outlier detection results of the two real data analyses, the ITK test has the same high power as the $T_k$ test statistic and lower error probabilities than the other six test statistics ($T_k$, $Z_k$, $N_k$, $D_k$, $V_k$, and $L_k$). In conclusion, compared with other test statistics, ITK has the highest detection capability for outliers and the lowest "false alarm". Thus, the ITK method is recommended to be used to identify multiple outliers in a sample.

In this paper, we design two algorithms based on the Monte Carlo and the kernel density estimation to obtain the critical values of $T_k$. How to derive the exact critical value of $T_k$ is an interesting problem. Further, in the case of $k$ being unknown, we take a conservative estimation of $k = \hat{k} = [\sqrt{n}]$. Thus, it is worth studying the problem of choosing a more appropriate value of $k$ in our ITK method. This article discusses only the case of multiple upper outliers existing in a gamma sample. Noting that lower outliers or both upper and lower outliers may exist in practice, it is necessary to extend our outlier detection methods to these situations. In addition, the masking effect with our methods is not discussed in this paper, which remains our future research. How to extend our approaches to other distributions is also an important topic.

**Author Contributions:** Conceptualization, X.L., T.W. and G.Z.; methodology, X.L.; software, X.L.; validation, X.L., T.W. and G.Z.; formal analysis, X.L. and T.W.; writing—original draft preparation, X.L.; writing—review and editing, X.L., T.W. and G.Z.; visualization, X.L. and T.W.; supervision, G.Z.; project administration, G.Z.; funding acquisition, G.Z. All authors have read and agreed to the published version of the manuscript.

**Funding:** This research was funded by the Beijing Natural Science Foundation (Grant No. Z210003).

**Data Availability Statement:** Open suorce. Data presented in the article can be obtained by visiting https://www.ons.gov.uk/ (accessed on 29 November 2022).

**Acknowledgments:** The authors are grateful to the anonymous referees for helpful comments and suggestions that greatly improved this paper.

**Conflicts of Interest:** The authors declare no conflict of interest.

## Appendix A

*Appendix A.1. Lemma 1 and Its Proof*

**Proof of Lemma 1.** For any $n > k$ ($k \neq 0$), we have $\frac{\max\limits_{k \neq n} X_k}{\sum_{j=1}^{n-1} X_j} = \max(\frac{X_1}{\sum_{j=1}^{n-1} X_j}, \cdots, \frac{X_{n-1}}{\sum_{j=1}^{n-1} X_j})$.

Thus, $\frac{\max\limits_{k \neq n} X_k}{\sum_{j=1}^{n-1} X_j}$ and $\frac{X_n}{\sum_{j=1}^{n-1} X_j}$ are independent if $(\frac{X_1}{\sum_{j=1}^{n-1} X_j}, \cdots, \frac{X_{n-1}}{\sum_{j=1}^{n-1} X_j})$ and $\frac{X_n}{\sum_{j=1}^{n-1} X_j}$ are inde-

pendent. Note that $\frac{X_{n-1}}{\sum_{j=1}^{n-1} X_j} = 1 - (\frac{X_1}{\sum_{j=1}^{n-1} X_j} + \cdots + \frac{X_{n-2}}{\sum_{j=1}^{n-1} X_j})$, so $\frac{\max\limits_{k \neq n} X_k}{\sum_{j=1}^{n-1} X_j}$ and $\frac{X_n}{\sum_{j=1}^{n-1} X_j}$ are

independent if $(\frac{X_1}{\sum_{j=1}^{n-1} X_j}, \cdots, \frac{X_{n-2}}{\sum_{j=1}^{n-1} X_j})$ is independent of $\frac{X_n}{\sum_{j=1}^{n-1} X_j}$. Observe that $X_1, \cdots, X_n$ are

independent and from gamma $(m, \theta)$, so the joint density of $(X_1, \cdots, X_n)$ is

$$f(x_1, \cdots, x_n) = \frac{1}{\Gamma^n(m)\theta^{nm}} e^{-\frac{\sum_{j=1}^{n} x_j}{\theta}} \prod_{j=1}^{n} x_j^{m-1}. \tag{A1}$$

Similar to [15], let $V_1 = \frac{X_1}{\sum_{j=1}^{n-1} X_j}, \cdots, V_{n-2} = \frac{X_{n-2}}{\sum_{j=1}^{n-1} X_j}, V_{n-1} = \frac{X_n}{\sum_{j=1}^{n-1} X_j}$, then the joint

density of $(V_1, \cdots, V_{n-2}, V_{n-1})$ is

$$f(v_1, \cdots, v_{n-2}, v_{n-1}) = \frac{\Gamma(nm)}{\Gamma^n(m)\theta^{nm}} [v_1 \cdots v_{n-2}(1 - (v_1 + \cdots + v_{n-2}))]^{m-1} v_{n-1}^{m-1}$$

$$\times (\frac{\theta}{1 + v_{n-1}})^{nm}, 0 < v_1, \cdots, v_{n-2} < 1, 0 < \sum_{j=1}^{n-2} v_j < 1, v_{n-1} > 0. \tag{A2}$$

It can be observed that the marginal densities of $(V_1, \ldots, V_{n-2})$ and $V_{n-1}$ are given by

$$f_1(v_1, \cdots, v_{n-2}) = \frac{\Gamma((n-1)m)}{\Gamma^{n-1}(m)} [(v_1 \cdots v_{n-2})(1 - (v_1 + \cdots + v_{n-2}))]^{m-1},$$

$$0 < v_1, \cdots, v_{n-2} < 1, 0 < \sum_{j=1}^{n-2} v_j < 1, \tag{A3}$$

and

$$f_2(v_{n-1}) = \frac{\Gamma(nm)}{\Gamma((n-1)m)\Gamma(m)\theta^{nm}} v_{n-1}^{m-1} (\frac{\theta}{1 + v_{n-1}})^{nm}, v_{n-1} > 0, \tag{A4}$$

respectively. Clearly, $f(v_1, \cdots, v_{n-2}, v_{n-1}) = f_1(v_1, \cdots, v_{n-2}) f_2(v_{n-1})$, so $(\frac{X_1}{\sum_{j=1}^{n-1} X_j}, \cdots,$

$\frac{X_{n-2}}{\sum_{j=1}^{n-1} X_j})$ is independent of $\frac{X_n}{\sum_{j=1}^{n-1} X_j}$. Therefore, Lemma 1 is proved. □

*Appendix A.2. Proofs of Theorems*

**Proof of Theorem 1.** Consider the null distribution defined by Equation (6), and distribution of the alternative model defined by Equation (7). The proof for Theorem 1 is an extension of that in [1], which discusses the single outlier detection in the exponential distribution. Suppose there are $n$ observations, denoted by $X_1, X_2, \cdots, X_n$, especially, $X_j$ ($j = n - k + 1, \cdots, n$) is an observation from the sample, which consists of the $k$ largest points. Therefore, the alternative hypothesis is

$$X_1, X_2, \cdots, X_{n-k} \sim f(x|m, \theta);$$

$$X_{n-k+1}, X_{n-k+2}, \cdots, X_n \sim f(x|m, \theta, \lambda). \tag{A5}$$

Denoting $T_k = \frac{\sum_{j=n-k+1}^{n} X_{(j)}}{\bar{X}}$ and $T = \frac{\sum_{j=n-k+1}^{n} X_j}{\bar{X}}$, we first prove that the test statistic $T$ is an MLR test statistic. Noting that under the $H$, $\{X_1, \cdots, X_n\}$ is a random sample from (6), the likelihood function is

$$\begin{aligned} L_H(m, \theta|x) &= \prod_{j=1}^{n} f(m, \theta|x_j) \\ &= \prod_{j=1}^{n} \frac{\theta^m}{\Gamma(m)} \cdot x_j^{m-1} \cdot e^{-\theta x_j} \\ &= \frac{\theta^{nm}}{\Gamma^n(m)} \cdot (\prod_{j=1}^{n} x_j^{m-1}) \cdot e^{-n\theta\bar{x}}. \end{aligned} \tag{A6}$$

Denote the associated log likelihood function as $\ln L_H(m, \theta|x) = mn \ln \theta - n \ln \Gamma(m) + (m-1) \sum_{j=1}^{n} \ln x_j - n\theta\bar{x}$, and let $\frac{\partial \ln L_H(m,\theta|x)}{\partial m} = -n\frac{\Gamma'(m)}{\Gamma(m)} + n \ln \theta + \sum_{j=1}^{n} \ln x_j = 0$, $\frac{\partial \ln L_H(m,\theta|x)}{\partial \theta} = \frac{nm}{\theta} - n\bar{x} = 0$, then we obtain the maximum likelihood estimates of $m$ and $\theta$, denoted by $\hat{m}$ and $\hat{\theta}$, i.e.,

$$\hat{\theta} = \frac{m}{\bar{x}}, \tag{A7}$$

and $\hat{m}$ satisfies $-n\frac{\Gamma'(m)}{\Gamma(m)} + n \ln m - n \ln \bar{x} + \sum_{j=1}^{n} \ln x_j = 0$; here, there is no explicit form solution for $\hat{m}$. The numerical value of $\hat{m}$ can be obtained by Newton-Raphson algorithm or extra-sample information. Therefore, if $m$ is known, and we substitue $\hat{\theta} = \frac{m}{\bar{x}}$ to $\ln L_H(m, \theta|x)$, then

$$\ln L_H(m, \hat{\theta}|x) = -n \ln \Gamma(m) - nm \ln \bar{x} + nm \ln m + (m-1) \sum_{j=1}^{n} \ln x_j - nm. \tag{A8}$$

Similarly, under the alternative hypothesis $\bar{H}$, we have

$$\begin{aligned} \ln L_{\bar{H}}(m, \hat{\theta}, \hat{\lambda}|x) = &- n \ln \Gamma(m) + nm \ln (n-k) + nm \ln m - nm \ln (n\bar{x} - \sum_{j=n-k+1}^{n} x_j) \\ &- nm + km \ln k - km \ln (n-k) + km \ln (\frac{n\bar{x} - \sum_{j=n-k+1}^{n} x_j}{\sum_{j=n-k+1}^{n} x_j}) \\ &+ (m-1) \sum_{j=1}^{n} \ln x_j. \end{aligned} \tag{A9}$$

Therefore, reject $H$ if $\frac{L_{\bar{H}}(m,\hat{\theta},\hat{\lambda}|x)}{L_H(m,\hat{\theta}|x)} \geq c$, i.e., $X_{n-k+1}, \cdots, X_n$ are outliers if $\ln L_{\bar{H}}(m, \hat{\theta}, \hat{\lambda}|x) - \ln L_H(m, \hat{\theta}|x) \geq \ln c$. Thus, we consider

$$\ln L_{\bar{H}}(m, \hat{\theta}, \hat{\lambda}|x) - \ln L_H(m, \hat{\theta}|x) = nm \ln \frac{n-k}{n-T} + km \ln (\frac{k}{n-k}(\frac{n-T}{T})), T > k. \tag{A10}$$

where $T = \frac{\sum_{j=n-k+1}^{n} X_j}{\bar{X}}$. Obviously, the derivative of Equation (A10) with respect to $T$ is

$$f(T) = \frac{nm}{n-T} - \frac{nkm}{(n-T)T}. \tag{A11}$$

It is clear that $f(T) > 0$ for $T > k$, and $\ln L_{\bar{H}}(m, \hat{\theta}, \hat{\lambda}|x) - \ln L_H(m, \hat{\theta}|x)$ about $T$ is monotone increasing. Thus, $T = \frac{\sum_{j=n-k+1}^{n} X_j}{\bar{X}}$ is an MLR test statistic; see [1,16,17].

In practice, it is too difficult to assure $X_j$ is not only the $j$th observation but also an observation from the $k$ largest observations. Therefore, to extend the $T$ test statistic for

the ordering the samples' situation, the multiple decision procedures will be used here. The null hypothesis remains unchanged, and the $i$th alternative hypothesis is

$$\bar{H}_i : X_{i1}, \cdots, X_{i,n-k} \text{ from } f(x|m,\theta);$$
$$X_{i,n-k+1}, \cdots, X_{i,n} \text{ from } f(x|m,\theta,\lambda). \tag{A12}$$

The number of such alternative hypotheses is $\binom{n}{k}$, and

$$\ln L_{\bar{H}_i}(m,\hat{\theta},\hat{\lambda}|x) - \ln L_H(m,\hat{\theta}|x) = nm\ln\frac{n-k}{n-T_i} + km\ln\left(\frac{k}{n-k}\left(\frac{n-T_i}{T_i}\right)\right), T_i > k, \tag{A13}$$

where $T_i = \frac{\sum_{j=n-k+1}^{n} X_{i,j}}{\bar{X}_i}$ with $\bar{X}_i = \frac{\sum_{j=1}^{n} X_{i,j}}{n}$. Subject to a probability of correct adoption of the null hypothesis $H$, the decision criterion is that of maximizing the power of adopting the correct $\bar{H}_i$. In the present situation of a gamma model, the multiple decision procedures lead to adopting $\bar{H}_i$ if $T_i$ is maximized and is sufficiently large. Because all observations are one-dimensional, outliers only exist at the upper end, and so the appropriate test statistic is

$$T_k = \frac{\sum_{j=n-k+1}^{n} X_{(j)}}{\bar{X}}. \tag{A14}$$

Theorem 1 is proved. $\square$

**Proof of Theorem 2.** Similar to [18], denote $a_n(v)$ and $A_n(v)$ as the density function and the cumulative distribution function (CDF) of $T_{n,(n)}$, respectively, and we have

$$a_n(v) = \lim_{dv \to 0} \frac{P(T_{n,(n)} \in (v, v+dv))}{dv}. \tag{A15}$$

Denote $\Omega = \bigcup_{j=1}^{n}\{T_{n,(n)} = T_{n,j}\}$, so

$$P(T_{n,(n)} \in (v, v+dv)) = P(\{T_{n,(n)} \in (v, v+dv)\} \bigcap \{\bigcup_{j=1}^{n}\{T_{n,(n)} = T_{n,j}\})$$

$$= P(\bigcup_{j=1}^{n}(T_{n,j} \in (v, v+dv), T_{n,(n)} = T_{n,j})). \tag{A16}$$

Note that $\{T_{n,j} \in (v, v+dv), T_{n,(n)} = T_{n,j}\}$ is incompatible with $\{T_{n,i} \in (v, v+dv), T_{n,(n)} = T_{n,i}\}$, for any $i \neq j$, thus, by the additivity of probability measures,

$$P(\bigcup_{j=1}^{n}(T_{n,j} \in (v, v+dv), T_{n,(n)} = T_{n,j}))$$

$$= \sum_{j=1}^{n} P(T_{n,j} \in (v, v+dv), T_{n,(n)} = T_{n,j})$$

$$= nP(T_{n,j} \in (v, v+dv), T_{n,(n)} = T_{n,j})$$

$$= nP(T_{n,n} \in (v, v+dv), T_{n,(n)} = T_{n,n})$$

$$= nP(T_{n,n} \in (v, v+dv), \max_{k \neq n} X_k < X_n)$$

$$= nP(T_{n,n} \in (v, v+dv), \max_{k \neq n} \frac{X_k}{\bar{X}_{[-n]}} < \frac{X_n}{\bar{X}_{[-n]}}), \tag{A17}$$

where $\bar{X}_{[-n]} = \frac{1}{n-1}\sum_{j=1}^{n-1} X_j$. Note that

$$
\begin{aligned}
T_{n,n} &= \frac{X_n}{\bar{X}} \\
&= n\frac{(n-1)\frac{X_n}{\sum_{j=1}^n X_j - X_n}}{(n-1)(1 + \frac{X_n}{\sum_{j=1}^n X_j - X_n})} \\
&= n\frac{\frac{X_n}{\bar{X}_{[-n]}}}{(n-1) + \frac{X_n}{\bar{X}_{[-n]}}}.
\end{aligned} \tag{A18}
$$

Since $X_1, \cdots, X_n$ are independent and from gamma $(m, \theta)$, $\frac{X_k}{\sum_{j=1}^{n-1} X_j}$ follows beta $(m, (n-2)m)$. Therefore,

$$
\begin{aligned}
(A17) &= nP(T_{n,n} \in (v, v+dv), T_{n-1,(n-1)} < \frac{X_n}{\bar{X}_{[-n]}}) \\
&= nP(T_{n,n} \in (v, v+dv), T_{n-1,(n-1)} < (n-1)\frac{T_{n,n}}{n - T_{n,n}}) \\
&= nP(T_{n,n} \in (v, v+dv))P(T_{n-1,(n-1)} < (n-1)\frac{T_{n,n}}{n - T_{n,n}}|T_{n,n} \in (v, v+dv)). \tag{A19}
\end{aligned}
$$

Because $T_{n-1,n} = n\frac{X_n}{\sum_{j=1}^{n-1} X_j}$ and $T_{n-1,(n-1)} = n\frac{\max_{k\neq n} X_k}{\sum_{j=1}^{n-1} X_j}$ are independent, we obtain

$$
\begin{aligned}
a_n(v) &= \lim_{dv\to 0}\{n\frac{P(T_{n,n} \in (v, v+dv))}{dv}P(T_{n-1,(n-1)} < (n-1)\frac{T_{n,n}}{n - T_{n,n}}|T_{n,n} \in (v, v+dv))\} \\
&= n\beta_{m,(n-1)m}(v)A_{n-1}[(n-1)\frac{v}{n-v}], \; 1 < v < n. \tag{A20}
\end{aligned}
$$

Theorem 2 is proved. $\quad\square$

## Appendix B

The appendix lists the alcohol-related mortality rates in selected countries in 2000 and artificial scout position data.

**Table A1.** Alcohol-related mortality rates in selected countries in 2000.

| Country | Mortality |
|---|---|
| Afghanistan | 0.01 |
| Algeria | 0.25 |
| Angola | 1.85 |
| Armenia | 2.90 |
| Australia | 10.17 |
| Austria | 13.2 |
| Azerbaijan | 0.65 |
| Bahrain | 2.15 |
| Bangladesh | 0.01 |
| Benin | 1.34 |
| Bhutan | 0.17 |
| Bolivia (Plurinational State of) | 2.32 |
| Brunei Darussalam | 0.37 |
| Cambodia | 1.51 |
| Central African Republic | 1.51 |
| Chad | 0.25 |
| Colombia | 4.66 |

**Table A1.** *Cont.*

| Country | Mortality |
|---|---|
| Comoros | 0.09 |
| Congo | 2.26 |
| Democratic Republic of the Congo | 1.98 |
| Denmark | 11.69 |
| Djibouti | 1.34 |
| Egypt | 0.14 |
| El Salvador | 2.79 |
| Eritrea | 0.83 |
| Estonia | 0.01 |
| Ethiopia | 0.88 |
| Fi Ji | 2.05 |
| France | 13.63 |
| Gambia | 2.18 |
| Ghana | 1.60 |
| Guatemala | 2.63 |
| Guinea | 0.17 |
| Guinea-Bissau | 2.84 |
| Honduras | 2.61 |
| Iceland | 6.17 |
| India | 0.93 |
| Indonesia | 0.06 |
| Iran | 0.01 |
| Iraq | 0.20 |
| Ireland | 14.07 |
| Israel | 2.53 |
| Jordan | 0.49 |
| Kenya | 1.51 |
| Kiribati | 0.46 |
| Kuwait | 0.01 |
| Kyrgyzstan | 2.13 |
| Lebanon | 2.26 |
| Libya | 0.01 |
| Madagascar | 1.16 |
| Malawi | 1.18 |
| Malaysia | 0.54 |
| Maldives | 1.83 |
| Mali | 0.47 |
| Mauritania | 0.03 |
| Mexico | 4.99 |
| Micronesia (Federated States of) | 2.23 |
| Mongolia | 2.79 |
| Montenegro | 0.01 |
| Morocco | 0.45 |
| Mozambique | 1.14 |
| Myanmar | 0.35 |
| Nepal | 0.08 |
| Niger | 0.1 |
| Oman | 0.38 |
| Pakistan | 0.02 |
| Papua New Guinea | 0.73 |
| Portugal | 11.89 |
| Qatar | 0.5 |
| Republic of Korea | 10.33 |
| Russian Federation | 10.18 |
| Samoa | 3 |
| Saudi Arabia | 0.05 |
| Senegal | 0.29 |
| Singapore | 2.03 |
| Slovenia | 11.9 |

**Table A1.** *Cont.*

| Country | Mortality |
|---|---|
| Solomon Islands | 0.71 |
| Somalia | 0.01 |
| Sri Lanka | 1.45 |
| Sudan | 1.76 |
| Syrian Arab Republic | 1.41 |
| Tajikistann | 0.37 |
| The former Yugoslav republic of Macedonia | 2.86 |
| Timor-Leste | 0.5 |
| Togo | 1.1 |
| Tonga | 1.24 |
| Tunisia | 1.21 |
| Turkey | 1.54 |
| Turkmenistan | 2.9 |
| United Arab Emirates | 1.64 |
| United Kingdom of Great Britain and Northern Ireland | 10.59 |
| Uzbekistan | 1.6 |
| Vanuatu | 1.21 |
| Viet Nam | 1.6 |
| Yemen | 0.07 |
| Zambia | 2.62 |
| Zimbabwe | 1.68 |

**Table A2.** Artificial scout position data.

| Soldier's ID | Position |
|---|---|
| 1 | 0.88 |
| 2 | 2.90 |
| 3 | 0.21 |
| 4 | 0.47 |
| 5 | 3.44 |
| 6 | 0.48 |
| 7 | 0.83 |
| 8 | 3.32 |
| 9 | 0.58 |
| 10 | 0.35 |
| 11 | 0.31 |
| 12 | 0.53 |
| 13 | 0.91 |
| 14 | 0.65 |
| 15 | 0.70 |
| 16 | 0.80 |
| 17 | 0.52 |
| 18 | 0.13 |
| 19 | 0.55 |
| 20 | 0.85 |

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
