# Peer review of "A Method for Detecting Outliers from the Gamma Distribution"

_axioms, doi:10.3390/axioms12020107_

Round 1

Reviewer 1 Report

Outliers often occur during the data collection, which could impact the result seriously and lead to a large inference error; therefore, it is important to detect outliers before data analysis. Gamma distribution is a popular distribution in statistics, this paper proposes a method for detecting multiple upper outliers in gamma(m, θ). For computing the critical value of the test statistic in proposed method, authors derive the density function for the case of a single outlier and design two algorithms based on the Monte Carlo and the kernel density estimation for the case of multiple upper outliers. A simulation study shows that the test statistic proposed in this paper outperforms some common test statistics. Finally, authors propose an improved testing method to reduce the impact of the swamping effect, which is demonstrated by real data analyses.

The paper is well written and discusses an interesting problem. Results are properly detailed. Everything looks consistent. And the mathematics seems correct to me. I list my comments as follows,

1. I would like to see a better motivation for their approach. Why exactly is it necessary? How does the improvement actually work?

2. More recent literature must be added and compared in relation to current work.

3. The results must be interpreted practically.

4. add more perspectives at the end of the conclusion.

5. The quality of the Figures is inferior, these figures must be saved in eps extension.

6. Correct the misprint

Reviewer 2 Report

It was an interesting paper to read and to review, due to its purpose and methods used for testing and validate the theorems proposed. The outliers problem is an important topic not only in the context of this journal, but for entire research field, because outilers occur during the data collection, which could impact the research result. In this context, it is important to detect outliers before data analysis. and to correct their influence on the results.We aăpreciate the use of density function for the case of a single outlier and the design of the two algorithms based on the Monte Carlo and the kernel density estimation for the case of multiple upper outliers. Also it was important to design a simulation study to show that the test statistic proposed in this paper outperforms some common test statistics. 

Reviewer 3 Report

File attached, I hope.

Round 2

Reviewer 1 Report

The authors have made appropriate corrections for all points.

I recommend publication in this journal.

Reviewer 3 Report

Attached

Round 3

Reviewer 3 Report

No further comments - all my corrections have been made. Any remaniing infelicities in English are negligible.